# The Highly Conservative Cysteine of Oncomodulin as a Feasible Redox Sensor

**DOI:** 10.3390/biom11010066

**Published:** 2021-01-06

**Authors:** Alisa A. Vologzhannikova, Polina A. Khorn, Marina P. Shevelyova, Alexei S. Kazakov, Victor I. Emelyanenko, Eugene A. Permyakov, Sergei E. Permyakov

**Affiliations:** Institute for Biological Instrumentation, Pushchino Scientific Center for Biological Research of the Russian Academy of Sciences, 142290 Pushchino, Russia; horn_pa@mail.ru (P.A.K.); marina.shevelyova@gmail.com (M.P.S.); fenixfly@yandex.ru (A.S.K.); emelyane@rambler.ru (V.I.E.); epermyak@yandex.ru (E.A.P.)

**Keywords:** calcium-binding protein, EF-hand, parvalbumin, oncomodulin, cysteine, thiol oxidation, disulfide dimerization, redox potential, redox sensor, protein stability

## Abstract

Oncomodulin (Ocm), or parvalbumin β, is an 11–12 kDa Ca^2+^-binding protein found inside and outside of vertebrate cells, which regulates numerous processes via poorly understood mechanisms. Ocm consists of two active Ca^2+^-specific domains of the EF-hand type (“helix-loop-helix” motif), covered by an EF-hand domain with inactive EF-hand loop, which contains a highly conservative cysteine with unknown function. In this study, we have explored peculiarities of the microenvironment of the conservative Cys18 of recombinant rat Ocm (rWT Ocm), redox properties of this residue, and structural/functional sensitivity of rWT Ocm to the homologous C18S substitution. We have found that *pK_a_* of the Cys18 thiol lays beyond the physiological pH range. The measurement of redox dependence of rWT Ocm thiol–disulfide equilibrium (glutathione redox pair) showed that redox potential of Cys18 for the metal-free and Ca^2+^-loaded protein is of −168 mV and −176 mV, respectively. Therefore, the conservative thiol of rWT Ocm is prone to disulfide dimerization under physiological redox conditions. The C18S substitution drastically reduces α-helices content of the metal-free and Mg^2+^-bound Ocm, increases solvent accessibility of its hydrophobic residues, eliminates the cooperative thermal transition in the apo-protein, suppresses Ca^2+^/Mg^2+^ affinity of the EF site, and accelerates Ca^2+^ dissociation from Ocm. The distinct structural and functional consequences of the minor structural modification of Cys18 indicate its possible redox sensory function. Since some other EF-hand proteins also contain a conservative redox-sensitive cysteine located in an inactive EF-hand loop, it is reasonable to suggest that in the course of evolution, some of the EF-hands attained redox sensitivity at the expense of the loss of their Ca^2+^ affinity.

## 1. Introduction

Oncomodulin (Ocm) or parvalbumin β (β-PA) is a small (11–12 kDa), acidic (pI < 5), globular vertebrate-specific Ca^2+^-binding protein of the EF-hand superfamily [1,2,3,4]. The EF-hand domain consists of a 12-residue Ca^2+^-coordinating loop flanked by two amphiphilic α-helices (“helix-loop-helix” motif) [5]. Ocm contains three EF-hand domains, named according to the flanking helices: AB, CD, and EF (see Appendix A). In contrast to the high-affinity CD and EF domains (their equilibrium Ca^2+^ dissociation constants reach 1 nM [6]), the truncated 10-residue loop of the AB domain (AB-loop) is inactive due to the lack of some critical residues (Appendix A). Meanwhile, the AB domain is essential for maintenance of structural stability of Ocm and its high affinity to Ca^2+^ [7,8].

The closest relative of Ocm, parvalbumin α (α-PA) [9], has similar structural properties [2,3]. While the CD and EF domain loops are the most conservative regions of both Ocm and α-PA, their AB domains exhibit relatively high variability, especially in the loop region (see Appendix A). The consensus sequences of AB-loops of Ocm and α-PA differ in 4 positions versus 2 and 1 positions for the CD and EF loops, respectively, indicating evolutionary specialization of Ocm/α-PA via modification of the AB-loop.

Despite structural similarities, Ocm and α-PA exhibit marked differences in their expression and functional profiles. Both parvalbumin (PA) isoforms were found in muscles in a species-specific manner, favoring muscle relaxation process (shown for α-PA [10,11]) and serving as major fish allergens (mostly Ocm; reviewed in [12]). While α-PA was found in both the outer and inner hair cells of cochlea, Ocm expression is restricted to the outer hair cells (2–3 mM) [13,14]. Targeted deletion of Ocm causes progressive cochlear dysfunction [15]. Ocm was found in placenta and in numerous cancers (reviewed in [16]), while α-PA is expressed in distal convoluted tubule [17]. Neuronal expression of Ocm is highly limited, and becomes noticeable in α-PA knockout mice [18]. Ocm exhibits hormone-like and growth factor activities (reviewed in [19]). It is expressed and secreted by macrophages and neutrophils and serves as a growth factor for neurons [20,21]. Chicken β-PA is expressed by epithelial cells in thymus and circulates in the blood, stimulating development of cell-mediated immunity (see in [19]). Ocm and α-PA are present in cutaneous mucus of some amphibians and fish genera, participating in bacterial defense via Ca^2+^ chelation and serving as chemoattractants for thamnophiine snakes [22]. In spite of the wide spectrum of biological activities of Ocm and α-PA, their targets and molecular mechanisms of action remain obscure. As Ca^2+^ binding induces only minor changes in tertiary structure of α-PA [23], it is considered as a Ca^2+^ buffer (as observed in muscles [10,11]). Meanwhile, the marked calcium sensitivity of tertiary structure of β-PA [24,25,26] argues that Ocm could serve as a Ca^2+^ sensor. Such activity was reported for rat Ocm, which recognizes and inhibits glutathione reductase under Ca^2+^ excess [27]. Importantly, Ca^2+^ binding induces in β-PA the structural rearrangements, which consistently involve the AB domain [24,25,26]. Therefore, the AB domain of Ocm might be crucial for Ca^2+^-dependent target recognition.

Positions 3, 5, 8, and 9 of the AB-loop are highly conservative in both Ocm and α-PA (see Appendix A). Positions 5, 8, and 9 of both PA lineages mainly contain Ala, Ser, and Phe residues, respectively. The only difference between the highly conservative residues of Ocm and α-PA is position 3, which mostly contains Cys in Ocm and Phe in α-PA. Therefore, the highly conservative Cys residue of Ocm could be critical for functional specialization of the protein.

Studies of reactivity of the conservative Cys of rat Ocm and functional consequences of its disulfide dimerization gave surprisingly conflicting results [28,29]. Meanwhile, direct structural data evidence exposure of the Cys18 to solvent upon Ca^2+^ removal [24] (Appendix A), in line with elevated reactivity of the apoprotein with Ellman’s reagent [28]. The analogous effect was reported for carp [30], perch [31], and silver hake [32] β-PAs. The exposure of Cys18 of rat Ocm is accompanied by disruption of the hydrogen bonds between carbonyl oxygen of Cys18 and side chain nitrogen atoms of Arg75, disappearance of the salt bridge between side chains of Arg75 and Glu81, along with formation of the hydrogen bond between side chains of Arg75 and Glu25 [24] (Appendix A). The evident Ca^2+^-induced changes in both solvent accessibility of Cys18 of rat Ocm and its interactions with neighboring groups point out that minor structural modifications of this residue could affect the Ca^2+^ binding process and/or structural properties of the Ca^2+^-bound/free protein states. To explore this possibility, we have studied influence of the homologous C18S substitution, serving as a model of the thiol oxidation, on conformational stability and metal-binding properties of recombinant rat Ocm. Our analysis of thiol–disulfide equilibrium of the Cys18 showed that its redox potential is close to the physiological level, which implies accumulation of disulfide dimer of Ocm under oxidizing conditions, as reported for some other EF-hand proteins. Overall, the data presented favor the view that the highly conservative Cys18 of rat Ocm likely serves as a redox sensor.

## 2. Materials and Methods

### 2.1. Materials

Molecular biology grade HEPES (N-(2-hydroxyethyl)piperazine-N′-(2-ethanesulfonic acid)), ultra-grade H_3_BO_3_/glycine/Tris (tris(hydroxymethyl)aminomethane), and MES (2-(N-morpholino)ethanesulfonic acid) were from Calbiochem (San Diego, CA, USA), Fluka (Munich, Germany), Sigma-Aldrich Co. (St. Louis, CA, USA), and Amresco (Solon, OH, USA), respectively. Pharma grade KCl, ultra-grade TCA (trichloroacetic acid), LC/MS grade acetonitrile, reagent-grade formic acid, and GSH (reduced L-glutathione)/GSSG (oxidized L-glutathione) were purchased from PanReac AppliChem. (Darmstadt, Germany). Biotechnology grade DTT (DL-dithiothreitol) was purchased from DiaM (Moscow, Russia). Biotechnology grade 2-ME (2-mercaptoethanol) and molecular mass markers for SDS-PAGE (sodium dodecyl sulfate polyacrylamide gel electrophoresis) were from Helicon (Moscow, Russia). Analytical grade bis-ANS (4,4′-dianilino-1,1′-binaphthyl-5,5′-disulfonic acid)/DTNB (5,5′-dithiobis(2-nitrobenzoic acid)), ultra-grade KOH/EDTA (ethylenediaminetetraacetic acid)/EGTA (ethylene glycol-bis(2-aminoethylether)-N,N,N′,N′-tetraacetic acid), standard solutions of CaCl_2_ and MgCl_2_ (Ca^2+^ content of 0.0005%) were purchased from Sigma-Aldrich Co. (St. Louis, CA, USA). Standard solutions of EDTA/EGTA potassium salt were prepared as described in [6]. Standard solution of Mg(NO_3_)_2_ was purchased from Fluka (Munich, Germany). Molecular biology grade glutaric aldehyde and ultra-grade CuCl_2_ were from Amersham Biosciences (Uppsala, Sweden) and Riedel-de Haën (Buchs, Switzerland), respectively. Biochemistry grade guanidinium chloride and Coomassie Brilliant Blue R-250 were products of Merck (Darmstadt. Germany). Toyopearl SuperQ-650M was purchased from Tosoh Bioscience (Griesheim, Germany). Sephadex G-25 and PD MidiTrap^TM^ G-25 were products of Pharmacia LKB (Uppsala, Sweden) and GE Healthcare (Chicago, IL, USA), respectively. d-10-camphorsulfonic acid was acquired from JASCO, Inc. (Tokyo, Japan). Other chemicals were reagent grade or higher.

All buffers and other solutions were prepared using ultrapure water (Millipore Simplicity 185 system). To avoid contamination of protein samples with Ca^2+^, only plastics or quartz ware were used instead of glassware. Degassed buffers were used for preparation of DTT solutions just before their usage to avoid DTT oxidation. Thermo SnakeSkin dialysis tubing (3.5 kDa MWCO) and Millipore Amicon Ultra centrifugal filters (3.0 kDa MWCO) were used for dialysis and concentration of protein solutions, respectively.

### 2.2. Preparation of Recombinant Rat Ocm

Recombinant wild type rat (*Rattus norvegicus*) oncomodulin (“rWT Ocm”) was isolated and purified as described earlier [33]. C18S substitution was introduced into rat Ocm by site-directed mutagenesis [34] of the plasmid encoding rWT Ocm via use of the oligonucleotide with mismatch in codon 18 (“TGC” triplet is replaced by “AGC”). C18S Ocm was expressed and purified similarly to rWT Ocm, but with the following modifications. The reducing agents were not used. The Toyopearl SuperQ-650M column (0.9 cm × 7.5 cm) was equilibrated with 20 mM Tris-HCl, 75 mM KCl, pH 8.5 (buffer B). The protein sample was dialyzed against buffer B and loaded onto the column. The latter was washed by buffer B and then by 20 mM Tris-HCl, 90 mM KCl, pH 8.5 buffer. The yield was 40–60 mg of protein per liter of cell culture for both Ocm samples. Purity of the protein samples was confirmed by native and SDS-PAGE, and characteristic UV absorption and fluorescence spectra (rat Ocm contains Tyr and Phe residues, but lacks Trp). The purified rWT Ocm samples were exhaustively dialyzed at 4 °C against distilled water with 1 mM 2-ME (avoided in the case of C18S Ocm), freeze dried, and stored at −20 °C. Before experiments, the dry rWT Ocm samples were dissolved in distilled water, reduced by addition of freshly prepared 5 mM DTT and purified from 2-ME and DTT by passage through PD MidiTrap G-25 column equilibrated with a buffer of choice. The reduced protein was used for experiments as soon as possible. Molecular masses of the Ocm samples were determined by liquid chromatography–mass spectrometry. The sample of 0.5 μM Ocm in 15:75 (*v/v*) mixture of deionized water and acetonitrile (10 mM formic acid) was passed through C18 sorbent (granule diameter of 2.7 µm and pore size of 300 Å), 15–100% acetonitrile gradient for 220 min, using EASY-nLC 1000 UHPLC chromatograph (Thermo Scientific, Waltham, MA, USA). The mass spectra were collected by Orbitrap Elite mass spectrometer (Thermo Scientific) in a positive ion mode (m/z from 600 to 2000) at detector voltage 1.8 kV, capillary temperature of 240 °C, and nebulizing nitrogen flow of 1 L/min. Ocm concentrations were calculated spectrophotometrically using the molar extinction coefficient at 280 nm, ε_280 nm_, derived from the molar extinction coefficients at 205 nm estimated according to the work in [35]: 2409 M^−1^cm^−1^ and 2721 M^−1^cm^−1^ for apo- and metal-loaded rWT Ocm, respectively; 2202 M^−1^cm^−1^ and 2575 M^−1^cm^−1^ for apo- and metal-loaded C18S Ocm, respectively.

### 2.3. Ellman’s Assay

Quantitation of thiol groups of rWT Ocm (1 Cys residue) was performed according to the modified Ellman’s test [36]. The dry rWT Ocm sample was dissolved in distilled water (100 μL), reduced by addition of freshly prepared 5 mM DTT and purified from 2-ME and DTT by passage through PD MidiTrap G-25 column equilibrated with 50 mM Tris-HCl, 1 mM EDTA, pH 8.0 buffer. rWT Ocm (20 µM) was unfolded by 6 M guanidinium chloride and treated by 1 mM DTNB for 10 min at 20 °C. The kinetics of the reaction was monitored by absorbance at 412 nm with a Cary 100 spectrophotometer (Varian Inc., Palo Alto, CA, USA) in a double-beam mode versus the reference solution without the protein. The molar extinction coefficient of 5-thio-2-nitrobenzoic acid at 412 nm of 13,700 M^−1^cm^−1^ (manual to Thermo Scientific cat. #22582) was used. Efficacy of the procedure used was confirmed by the analogous examination of reduced L-glutathione solution.

### 2.4. Removal of Metal Ions from Ocm

The contaminating Ca^2+^/Mg^2+^ ions were removed from rWT Ocm using precipitation by TCA [37] of the preliminary reduced protein (see above) (10–50 mM HEPES-KOH with/without 100 mM KCl, pH 7.0–8.2 or 10-20 mM H_3_BO_3_-KOH, pH 8.4–8.8 or 20 mM glycine-KOH, pH 9.2 or 10 mM Tris-HCl, pH 7.3–7.4 buffer), followed by purification from Ca^2+^/Mg^2+^ and TCA by gel filtration method [38] using Sephadex G-25 column equilibrated with a respective buffer. To protect the reduced rWT Ocm from oxidation, 5 mM of freshly prepared DTT solution was added prior to the precipitation by TCA and before the gel filtration. In case of C18S Ocm, the dry protein sample was dissolved in the same buffer and purified from metal ions in the identical manner, but without addition of DTT. This procedure was used for metal removal from rWT/C18S Ocm for preparation of disulfide dimer of rWT Ocm and its redox studies, CD measurements of apo- and Mg^2+^-loaded Ocm, DSC measurements of Mg^2+^-loaded Ocm, chemical cross-linking and DLS measurements, bis-ANS fluorescence, and metal affinity studies. Only the gel filtration method [38] was used for spectrophotometric pH titrations of Ocm, CD measurements of Ca^2+^-loaded Ocm, DSC measurements of apo- and Ca^2+^-loaded Ocm, and kinetic studies of metal association/dissociation.

### 2.5. Spectrophotometric pH Titration of Ocm

Estimation of *pK_a_* value of Cys18 of Ocm at 20 °C was carried out spectrophotometrically mainly as described in ref. [39]. UV absorption spectra of rWT/C18S Ocm (23 μM) were measured with a Cary 100 spectrophotometer (Varian Inc., Palo Alto, CA, USA), equipped with NESLAB RTE-7 circulator (Thermo Scientific, Waltham, MA, USA). Buffer conditions: 30 mM MES, 30 mM HEPES, 30 mM H_3_BO_3_, 30 mM glycine, 100 mM KCl, 1.5 mM EDTA (apo-Ocm), or 1 mM CaCl_2_ (Ca^2+^-bound Ocm). The protein solution was titrated by small aliquots of KOH stock solution. To avoid solution contamination with Ca^2+^ during the titrations, pH values were independently measured under identical conditions. The protein absorbance at 240 nm was corrected for a minor light scattering contribution via extrapolation with power function of the long wavelength region of Ocm absorption spectrum. Contribution of deprotonated Tyr residues to the absorbance at alkaline pH values [40] was subtracted from the experimental data.

### 2.6. Preparation of Disulfide Dimer of rWT Ocm

Disulfide dimer of rWT Ocm (“dOcm”) was prepared by Cu^2+^-induced oxidation of the protein reported earlier [28]. Two-hundred micromolar CuCl_2_ was added to the metal-depleted rWT Ocm (50 μM) in 50 mM HEPES-KOH, 100 mM KCl, pH 7.4 buffer (refer to Removal of metal ions from Ocm section), followed by incubation at 25 °C for 24 h. Metal ions were removed from the protein sample using the same metal removal procedure (TCA treatment, followed by gel filtration), but without DTT usage. dOcm solution was frozen and stored at −20 °C. The SDS-PAGE analysis of the sample under non-reducing/reducing conditions confirmed total conversion of monomeric rWT Ocm into the disulfide dimer (data not shown).

### 2.7. Redox Dependence of Disulfide Dimerization of rWT Ocm

The redox potential of Cys18 of rWT Ocm was measured mainly as described in [41]. Metal-free/Ca^2+^-loaded dOcm solution (1.5 μM) in 50 mM HEPES-KOH, 100 mM KCl, 1 mM EDTA/CaCl_2_, pH 7.4 buffer was incubated with 0.2 mM GSSG and various concentrations of GSH (25 μM to 1.6 mM). The dissolved oxygen was removed from the solutions using a degassing station from TA Instruments (pressure of 25 mm Hg for 20 min, stirring at 700 rpm), followed by saturation of the solutions with argon by bubbling 99.993% argon for 7–10 min. The reaction solutions were sealed under argon, incubated at 35 °C for 62 h, and analyzed by non-reducing SDS-PAGE (15% resolving gel). The gels were stained with Coomassie Brilliant Blue R-250 and scanned using Molecular Imager PharosFX Plus System (Bio-Rad Laboratories, Inc., Hercules, CA, USA). The weight fractions of Ocm forms were calculated by Quantity One software (Bio-Rad Laboratories, Inc., Hercules, CA, USA).

The redox equilibrium in the system of a protein with one thiol group and GSH/GSSG couple can be described by the scheme [I]:*K**M*–*M* + 2 *GSH* ↔ 2 *M* + *GSSG*,
[I]
where *M–M* and *M* denote disulfide dimer and monomer of the protein, respectively, and *K* is an apparent dimer dissociation constant (*K* = [*M*]^2^[*GSSG*]/[*M-M*][*GSH*]^2^). Combining the equation for *K* with the material balance equation 2[*M–M*] + [*M*] = *M*_0_ (*M_o_* is a total concentration of the protein monomer) it is easy to obtain weight fraction of the dimeric protein, *f_D_*, which is expressed as follows,
(1)$$f_{D}=\frac{2\left[M-M\right]}{M_{0}}=\frac{K•10^{x}+4M_{0}-\sqrt{{\left(K•10^{x}\right)}^{2}+8M_{0}K•10^{x}}}{4M_{0}}$$
where *x* equals to *log*([*GSH*]^2^/[*GSSG*]).

The experimental dependence of *f_D_* on *x* for dOcm sample was fitted by Equation (1) using OriginPro v.9.0 (OriginLab Corp., Northampton, MA, USA) software and *K* as a fitting parameter (the changes in [*GSH*] and [*GSSG*] due to reaction [I] were neglected).

Considering that at the middle of the transition between *M*–*M* and *M* (scheme [I]) their weight fractions are equal, the *x* value at this point can be expressed as follows.
(2)$$x_{1/2}=\log\frac{M_{0}}{K}$$

At pH below *pK_a_* value of GSH (8.65 at 25 °C and ionic strength, *I*, of 0.1 M [42]), the equilibrium between GSSG and GSH is described by the following scheme [43].
*GSSG* + 2 *H^+^* + 2 *e^−^* ↔ 2 *GSH*[II]

The respective redox potential of the GSSG/2GSH redox pair, *E_h_*, is defined by Nernst equation (activity coefficients of GSSG and GSH are commonly assumed to be close to 1) [42]:(3)$$E_{h}=E_{0}-\frac{RT}{2F}\ln\frac{{\left[GSH\right]}^{2}}{\left[GSSG\right]}=E_{0}-\frac{RT}{2F}\ln10•x$$
where *E_o_* is a standard reduction potential for the GSSG/2GSH couple and *R*, *T*, and *F* are gas constant, absolute temperature, and Faraday constant, respectively. The *E_o_* value at the experimental conditions used here (pH 7.4 and *I* of 0.12 M) was estimated by linear interpolation of the *E_o_* values at pH 7 and 8 and *I* of 0 and 0.25 M (25 °C [44]), which gives −297 mV. Thus, at 35 °C Equation (3) can be rewritten as
(4)$$E_{h}=-297mV-30.6mV•x$$

The experimental *x* values were converted into the scale of redox potential of the GSSG/2GSH redox pair using Equation (4).

### 2.8. Circular Dichroism (CD) Measurements

Far-UV CD studies were carried out at 20 °C using J-810 spectropolarimeter (JASCO, Inc.), equipped with a Peltier-controlled cell holder. The instrument was calibrated with an aqueous solution of d-10-camphorsulfonic acid and purged with nitrogen. The quartz cell with pathlength of 1 mm was used. Ocm concentration was 7–13 µM. Buffer conditions: 10 mM H_3_BO_3_-KOH, pH 8.8, 1.5 mM EDTA or 1 mM CaCl_2_ for apo- and Ca^2+^-loaded Ocm, respectively, and 10 mM Tris-HCl, pH 7.3 with 1 mM EGTA, 1 mM MgCl_2_ for Mg^2+^-loaded Ocm. The spectral contribution of the buffer was subtracted from the experimental spectra. Bandwidth was 2 nm, averaging time 2 s, and accumulation 3. Quantitative estimates of the secondary structure fractions were performed using CDPro software [45]. The experimental data in the 200–240 nm range were treated by SELCON3, CDSSTR, and CONTIN algorithms, using SDP48 and SMP56 reference protein sets. The final secondary structure fractions represent averaged values.

### 2.9. Chemical Cross-Linking Experiments

The metal-depleted rWT/C18S Ocm (1.6–1.7 mg/mL) was cross-linked with 0.02% glutaric aldehyde at 20 °C in 20 mM H_3_BO_3_-KOH, pH 8.4 buffer, 1.5 mM EDTA or 1 mM CaCl_2_/MgCl_2_. The reaction proceeded overnight and was terminated with a SDS sample buffer. The samples were subjected to reducing SDS-PAGE (5% concentrating and 15% resolving gels; 5 µg of Ocm per lane) and staining with Coomassie Brilliant Blue R-250. The gel was scanned using Molecular Imager PharosFX Plus System and analyzed by Quantity One software (Bio-Rad Laboratories, Inc.).

### 2.10. Dynamic Light Scattering Studies (DLS)

DLS measurements were carried out for rWT/C18S Ocm at 20.0 °C using a Zetasizer Nano ZS system (Malvern Instruments Ltd.). The backscattered light from a 4 mW He-Ne laser (632.8 nm) was collected at an angle of 173°. Protein concentration was 1.3–1.7 mg/mL. The buffer conditions: 20 mM H_3_BO_3_-KOH, pH 8.4, 1.5 mM EDTA or 1 mM CaCl_2_/MgCl_2_. The protein samples were preliminarily passed through 0.02 µM Whatman Anotop 10 syringe filters. The acquisition time for a single autocorrelation function was 100 s. The resulting autocorrelation function was an average of ten measurements. The intensity-weighted size distributions were calculated using preset parameters for distilled water at 20 °C (refractive index of 1.33 and a viscosity value of 1.0031 cP). The estimates of mean hydrodynamic radii (*R_h_*) of Ocm were derived from these distributions.

### 2.11. The Scanning Calorimetry Measurements

The differential scanning calorimetry (DSC) studies were carried out on a Nano DSC microcalorimeter (TA Instruments) at a heating rate of 1 K/min and an excess pressure of 4 bar. Ocm concentration was 1.2–2.1 mg/mL. Buffer conditions: 10 mM H_3_BO_3_-KOH, 1 mM CaCl_2_, pH 9.0 (Ca^2+^-loaded Ocm); 20 mM glycine-KOH, 1 mM EDTA, pH 9.2 (apo-Ocm) or 20 mM glycine-KOH, 1 mM MgCl_2_, pH 9.2 (Mg^2+^-loaded Ocm). The DSC measurements and calculations of protein specific heat capacity (*C_p_*) were performed as described earlier [46]. The partial molar volume of Ocm and specific heat capacity of the fully unfolded protein were estimated according to the works in [47,48], respectively. The temperature dependence of *C_p_* was analyzed according to the cooperative two-state model:*n*∙N ↔ *n*∙D
[III]
where N and D denote native and denatured protein states, respectively, and *n* is a number of molecules involved in the transition. The experimental data were fitted using OriginPro v.9.0 software (OriginLab Corp., Northampton, MA, USA), using the equations derived earlier [49]. The heat capacity change accompanying the transition (∆*C_p_*) was supposed to be independent of temperature. ∆*C_p_*, mid-transition temperature (*T*_0_), enthalpy of protein denaturation at temperature *T*_0_ (∆*H*_0_) and *n* were used as fitting parameters.

### 2.12. Fluorescence Measurements

Fluorescence studies were carried out at 20 °C on a Cary Eclipse spectrofluorimeter (Varian, Inc.), equipped with a Peltier-controlled cell holder. Quartz cells with pathlength of 10 mm were used. Protein and bis-ANS fluorescence were excited at 275 nm and 385 nm, respectively. Maximum intensities (*I_max_*) and maximum positions (*λ_max_*) of fluorescence emission spectra were obtained from fits of the spectra with log-normal curves [50] using LogNormal software (IBI RAS, Pushchino).

### 2.13. Fluorimetric Estimation of Ca^2+^ Affinity of Ocm

Ca^2+^ affinity of rWT/C18S Ocm at 20 °C was estimated mainly as described in [6] from spectrofluorimetric titration of the metal-depleted protein with CaCl_2_ standard solution, followed by titration of the Ca^2+^-loaded protein with EDTA potassium salt standard solution. Ocm concentration was 20–23 μM. Buffer conditions: 10 mM HEPES-KOH, pH 8.2. Intrinsic Ocm fluorescence intensities were corrected for the dilution effect via division of the experimental values by a factor of (1–10^−*Dexc*^), where *D_exc_* is the protein absorption at the excitation wavelength [51]. The experimental data were described by the sequential metal binding scheme [2]:*K*_*a*1_P + M ↔ P∙M,*K*_*a*2_P∙M + M ↔ P∙M_2_,
[IV]
where P and M denote protein and metal ion, respectively, and *K_a_*_1_ and *K_a_*_2_ are equilibrium metal association constants for the two active EF-hands of Ocm. The competition between Ocm and EDTA for metal ions was taken into account:*K_EDTA_*
EDTA + M ↔ EDTA∙M
[V]

The metal association constant for EDTA, *K**_EDTA_*, was calculated according to the work in [52], using the thermodynamic constants derived from IUPAC Stability Constants Database, SC-Database v.4.79 (Academic Software, Timble, UK), and corrected for temperature and ionic strength using van’t Hoff and Davies equations, respectively. The data were globally fitted according to the schemes [IV] and [V] using FluoTitr v.1.4 software (IBI RAS, Pushchino, Russia). Log(*K_a_*_1_), log(*K_a_*_2_) and specific fluorescence intensities of Ocm states were used as the fitting parameters. *K_a_*_1_ and *K_a_*_2_ values were estimated at 295 nm, 306 nm and 315 nm, and averaged.

### 2.14. Estimation of Kinetic Metal Association/Dissociation Constants

Stopped-flow fluorescence measurements were performed using RX.2000 Rapid Mix Accessory (Applied Photophysics Ltd., Leatherhead, UK) with a dead time of 6 ms. Temperature of all solutions was kept at 20 °C using NESLAB RTE-7 circulator (Thermo Scientific) and Peltier-controlled cell holder. Equal volumes (0.3 mL) of 5–18 µM rWT/C18S Ocm (Ca^2+^ to protein molar ratio of 2:1, or 5:1 for Mg^2+^ and the metal-depleted protein) and 1–2 mM EGTA/EDTA in 10 mM HEPES-KOH buffer, pH 8.2 (Ca^2+^ dissociation experiments) or pH 7.3 (Mg^2+^ dissociation), were mixed using a pneumatic drive. To prevent Ca^2+^ contamination of Ocm solution during the Mg^2+^ dissociation studies, 1 mM EGTA was added. Tyr fluorescence emission was measured at 305 nm. Each experiment was repeated 4 times. The experimental data were fitted within the sequential metal binding scheme [IV] [2] as described earlier [53].

### 2.15. Estimation of Mg^2+^ Affinity of Ocm by Equilibrium Dialysis

Mg^2+^ binding to rWT/C18S Ocm was studied by equilibrium dialysis method using a 96-well micro-equilibrium dialysis system (HTDialysis, LLC) [54,55]. Each well (500 μL) of the Teflon block is separated by dialysis membrane (regenerated cellulose with MWCO of 3.5 kDa, preliminarily soaked in 20% ethanol). The membrane and the block were cleaned from metal ions according to the procedure described in [56]. One half of the each well was filled with 200 μL of 30–220 µM solution of Ocm in a buffer (30 mM HEPES-KOH, 1 mM EGTA, pH 7.4), whereas the another half contained 200 μL of the same buffer with 5–1500 µM Mg(NO_3_)_2_ without Ocm. The wells were tightly sealed and equilibrated by continuous shaking (130 rpm) of the block at (20.0 ± 0.5) °C for 17–20 h. Total concentrations of Mg^2+^ in the equilibrated solutions were measured by electrothermal atomization atomic absorption spectrometer iCE 3000 (Thermo Scientific), using argon as an inert gas. The magnesium absorption band at 202.6 nm and Zeeman background correction were used. Ten microliters of diluted sample solution was analyzed in the graphite cuvette from 2 to 3 times. The analytical signal was calibrated using Mg(NO_3_)_2_ standard solution (0–40 μg/L) in the same buffer. Concentration of Mg^2+^ bound to Ocm ([bM]) was estimated for the each well as a difference between the total Mg^2+^ concentrations measured for both halves of the well, assuming that free Mg^2+^ concentrations ([fM]) do not differ between the two halves of the well. The total protein concentration (P_0_) was determined spectrophotometrically for the each well after the equilibrium dialysis. Since at Ocm concentration of 220 μM the Donnan coefficient is close to 1, the error arising due to the Gibbs–Donnan effect is negligible compared to the experimental uncertainties in Mg^2+^ and Ocm concentrations. The experimental data were analyzed using the sequential metal binding scheme [IV]. The number of Mg^2+^ ions bound per Ocm molecule was described by the following equation.
(5)$$\frac{\left[\mathrm{bM}\right]}{P_{0}}=\frac{K_{1}[\mathrm{fM}]+2\cdot K_{1}K_{2}{\left[\mathrm{fM}\right]}^{2}}{1+K_{1}[\mathrm{fM}]+K_{1}K_{2}{\left[\mathrm{fM}\right]}^{2}}$$

Log(*K*_1_) and log(*K*_2_) were used as fitting parameters in OriginPro v.9.0 software (OriginLab Corp., Northampton, MA, USA). The weighting factor $w_{i}=\frac{1}{\sigma_{i}^{2}}$ was used for the experimental points.

## 3. Results and Discussion

### 3.1. Isolation of Recombinant Rat Ocm

Recombinant wild type rat (*Rattus norvegicus*) oncomodulin (rWT Ocm; Swiss-Prot entry P02631) and its mutant form with homologous substitution of the highly conservative Cys18 in the inactive AB-loop (Appendix A) for Ser (C18S Ocm) were isolated mainly as described in [33]. The protein samples were homogeneous as judged by native and SDS-PAGE. The absence of Trp-containing contaminating proteins in the Ocm samples was confirmed by the lack of Trp contribution in their UV absorption and fluorescence spectra (data not shown). The rWT Ocm sample was stored in the presence of 2-ME to prevent unfavorable oxidation of Cys18 thiol. The 2-ME molecules were removed from the protein by fresh DTT treatment, followed by Ocm cleaning from the reducing agents by gel filtration (see Materials and methods section). The resulting free thiol content was 0.89 thiols per Ocm molecule, as evidenced by Ellman’s test.

Mass spectrometry measurements revealed two major fractions of the reduced rWT Ocm sample. One of them (~45%) had a mass of 12,057 Da, which is consistent with mass of the reduced Swiss-Prot entry P02631 without Met1 and N-terminal acetyl group. The other rWT Ocm fraction (~46%) was 131 Da heavier, in line with the presence of Met1. The minor fractions (~9%) of this sample corresponded to the Ca^2+^-bound and/or oxidized states of the abovementioned forms. The major fraction (~81%) of C18S Ocm had a mass of 12,041 Da, which corresponds to the mass expected for the protein without Met1 and N-terminal acetyl group. The remaining C18S Ocm (~19%) had a mass consistent with the presence of Met1.

### 3.2. Thiol Group pK_a_ Value

As the reactivity of a thiol group depends on its *pK_a_* value, we tried to estimate the *pK_a_* value of Cys18 of rWT Ocm. Deprotonated alkyl thiol group displays an absorption band around 240 nm (ε ≈ 4000 M^−1^cm^−1^) in the UV absorption spectrum, where absorption of the protonated thiol is negligible [57]. Therefore, the pH-induced change in ionic state (protonation-deprotonation) of the single thiol of rWT Ocm can be monitored by the protein absorbance at 240 nm, after subtraction of spectral contribution of deprotonated Tyr residues [40] of Ocm. Surprisingly, the resulting pH dependence of molar extinction coefficient at 240 nm (ε_240 nm_) for apo- and Ca^2+^-loaded forms of reduced rWT Ocm showed no changes within pH range from 6.5 to 9.5 (Figure 1). The identical control experiments for C18S Ocm gave analogous results. This means that the the single thiol of rWT Ocm does not change its ionic state within the pH range from 6.5 to 9. The changes in ε_240 nm_ at pH above 9.5 can be related to Cys18 deprotonation and/or alkaline denaturation of the protein. As average *pK_a_* value of Cys in folded proteins is 6.8 ± 2.7 [58], the result obtained show that Cys18 of rWT Ocm is characterized by *pK_a_* value which is either well below the average level or above most of the values reported for Cys residues of folded proteins, regardless of calcium level. In any case, the Cys18 of rWT Ocm is expected to be either fully protonated (*pK_a_* > 9.5) or fully deprotonated (*pK_a_* < 6.5) at physiological pH values, in contrast to the conservative Cys38 of the inactive EF-hand loop of bovine recoverin (*pK_a_* = 7.6 [39]). As Ca^2+^ removal from Ocm increases solvent accessibility of the conservative Cys residue [24,28,30,31,32], the thiol group of apo-Ocm is expected to exhibit a higher *pK_a_* value compared to that for the Ca^2+^-bound protein. Meanwhile, we did not detect this effect.

### 3.3. Redox Potential of Cys18 of rWT Ocm

The high intracellular concentrations of Ocm (up to 2–3 mM [14]) favor its involvement into thiol-disulfide exchange reactions. One of them is the interaction with a major cell redox buffer, glutathione (GSH). The studies of thiol-disulfide exchange between rWT Ocm and GSH were performed with disulfide dimer of the protein (referred to as “dOcm”), which was prepared by Cu^2+^-induced oxidation of rWT Ocm (see Preparation of disulfide dimer *of rWT Ocm* section). The equilibrium between metal-free/Ca^2+^-loaded dOcm (1.5 μM) and glutathione redox couple (scheme [I]) at 35 °C has been studied mainly as described in [41], in the presence of 0.2 mM GSSG and 25 μM–1.6 mM GSH, which corresponds to the redox potential of GSSG/2GSH redox pair (scheme [II]), *E_h_*, from −128 mV to −239 mV (Equation (4)). The incubation of this system under anaerobic conditions for 62 h ensures its efficient equilibration [41]. The weight fractions of dOcm, *f_D_*, were estimated using non-reducing SDS-PAGE (Figure 2). Regardless of Ca^2+^ content, the more reducing solvent conditions induce dOcm reduction with half-transition at *E_h_* values about −170 mV. The experimental data are well described by the simplest scheme [I] (Equation (1); Figure 2), which gives the apparent dimer dissociation constant, *K*, of 0.026 ± 0.004 and 0.048 ± 0.003, for Ca^2+^-loaded and apo-Ocm, respectively. The corresponding *x_1/2_* values are −3.94 ± 0.06 and −4.20 ± 0.03 (Equation (2)), while the *E_h_* values are (−176.5 ± 1.9) mV and (−168.4 ± 0.9) mV (Equation (4)), for Ca^2+^-loaded and apo-Ocm, respectively. The resulting *E_h_* values are close to those reported for erythrocytes of diabetic patients (−179 mV [59]), apoptotic murine hybridoma and HL-60 cells (−170 mV and −167 mV, respectively), and differentiating HT29 cells (−160 mV) (reviewed in [43]). Therefore, the intracellular conditions accompanied by the elevated redox potential are likely to favor accumulation of disulfide dimer of Ocm. Furthermore, extracellular Ocm [19,20,21] is confronted with even more oxidizing conditions with the *E_h_* values of −140 mV to −131 mV, which drastically increase with age (−93 mV to −84 mV) and under such pathological conditions as diabetes and age-related macular degeneration (up to −65 mV) [59]. Early atherosclerosis [60] and smoking [61] are accompanied by less marked increase in the *E_h_* values. The disulfide dimer of Ocm accumulated under the more oxidizing conditions is expected to possess an altered functional activity, as exemplified by efficient activation by the dimer of two calmodulin-dependent enzymes [29]. As Ocm regulates GSSG and GSH levels via Ca^2+^-dependent inhibition of glutathione reductase [62], the disulfide dimerization of Ocm could affect the intracellular redox potential due to the respective changes in the [GSH]/[GSSG] ratio.

The difference in *pK_a_* values of the conservative thiols located in the inactive EF-hand loops of rat Ocm and bovine recoverin (see above) is probably translated into differences in their *E_h_* values (−152 and −128 mV, for the Ca^2+^-loaded and Ca^2+^-free recoverin [41]). The relatively high *E_h_* values of recoverin are sufficient for manifestation of its disulfide dimerization under in vivo conditions [41,63], thereby indicating such possibility for Ocm. The *E_h_* values decrease upon calcium binding to Ocm and recoverin, implying that calcium promotes accumulation of their disulfide dimers.

### 3.4. Impact of C18S Substitution on Structural and Functional Properties of Rat Ocm

Aside from disulfide dimerization of Ocm, oxidizing physiological conditions may induce numerous chemical modifications of its Cys18 residue, including formation of mixed disulfides, sulfenic, sulfinic, and sulfonic acids [64], as observed for recoverin [63]. To explore the susceptibility of rWT Ocm properties to minor structural modifications of the Cys18, the latter was replaced by a homologous residue, Ser.

The secondary structure fractions for apo- and Mg^2+^/Ca^2+^-loaded rWT/C18S Ocm were estimated by far-UV CD using CDPro software [45] (Table 1). Although α-helices content of Ca^2+^-loaded rWT Ocm (55%) is in line with the X-ray structure 1OMD (55.6%), α-helicity of the apoprotein (32%) is 22% lower as compared to that derived from the NMR (nuclear magnetic resonance) structure 2NLN (54.6%). The latter is likely due to the Na^+^-binding induced stabilization of Ocm structure during NMR measurements (150 mM NaCl should favor Na^+^ binding to apo-Ocm according to its dissociation constant of 17 mM [65]). While Ca^2+^-bound states of rWT and C18S Ocm exhibit equivalent secondary structures, α-helices content of Mg^2+^-bound and metal-free states of C18S Ocm are lowered compared to rWT Ocm by 20–21%, partly at expense of the increase in β-sheets content (Table 1). Overall, secondary structure of apo-C18S Ocm is severely disorganized, but recovers upon Ca^2+^ binding, while Mg^2+^ binding is insufficient for full recovery of the secondary structure. The disturbance in secondary structure of rat Ocm in response to C18S substitution increases solvent exposure of its hydrophobic residues, as judged from elevated fluorescence intensity of a hydrophobic fluorescent probe 4,4′-dianilino-1,1′-binaphthyl-5,5′-disulfonic acid (bis-ANS), regardless of Mg^2+^/Ca^2+^ content (Table 1).

As the increased solvent exposure of hydrophobic residues of C18S Ocm could promote its multimerization, the quaternary structure of rWT/C18S Ocm was studied by chemical cross-linking and dynamic light scattering (DLS) at 20 °C (Table 2). SDS-PAGE of the Ocm forms cross-linked with 0.02% glutaric aldehyde shows that vast majority of the protein is monomeric, regardless of Mg^2+^/Ca^2+^ content. Therefore, the increase in mean hydrodynamic radius (*R_h_*) of rWT/C18S Ocm observed upon the metal depletion and Ca^2+^ replacement by Mg^2+^ reflects expansion of the protein molecule due to disorganization of its structure, in accordance with the CD data (Table 1). While *R_h_* values of the Mg^2+^/Ca^2+^-loaded rWT and C18S Ocm do not differ from each other, apo-state of C18S Ocm is considerably less compact compared to apo-state of rWT Ocm, in line with the drastically lowered α-helicity of the mutant.

As metal-free C18S Ocm exhibits the combination of features specific for the proteins lacking rigid tertiary structure, we have studied the integrity of its structure by differential scanning calorimetry, DSC (Figure 3). In contrast to the distinct heat sorption peak observed for apo-state of rWT Ocm (mid-transition temperature, *T*_0_, estimated within the cooperative two-state model [III], of 30.7 °C–Table 3), metal-free C18S Ocm does not reveal measurable heat sorption peaks characteristic for first-order thermal transitions. Furthermore, its specific heat capacity closely approaches that calculated for the fully unfolded protein [48]. Therefore, DSC data also strongly evidence unordered tertiary structure in apo-state of C18S Ocm. Meanwhile, Mg^2+^/Ca^2+^-loaded states of C18S Ocm demonstrate well-defined heat sorption peaks, shifted relative to the corresponding peaks of rWT Ocm towards lower temperatures by 5 °C and 3.7 °C, respectively (Table 3). Thus, Mg^2+^/Ca^2+^ binding evens out the differences between the properties of rWT and C18S Ocms, as observed for hydrodynamic radius (Table 2) and α-helices content (Table 1).

The drastic changes in structural properties of metal-depleted Ocm induced by C18S substitution could affect kinetic and equilibrium parameters of Ca^2+^/Mg^2+^-binding to the protein, which are crucial for PA functioning (at least as a muscle relaxation factor [10,11]). Ca^2+^-binding to rWT/C18S Ocm was explored using spectrofluorimetric titrations of their apo-forms by Ca^2+^, followed by competitive Ca^2+^ removal with Ca^2+^ chelator, EDTA (Figure 4a). The binding of Ca^2+^ to the proteins results in an increase in Tyr fluorescence intensity (rat Ocm has no Trp residues), while the Ca^2+^ removal causes the opposite effect. Analysis of the data according to the sequential metal-binding scheme [IV] shows that our estimates of Ca^2+^ affinity of rWT Ocm (Table 4) exceed the previous estimates [66] by 1–1.6 orders of magnitude, which is likely due to much lower KCl concentration used in our work. It is clearly seen that Ca^2+^ ions are removed from C18S Ocm by EDTA essentially easier than from rWT Ocm, implying that the latter has higher calcium affinity (Figure 4a). Indeed, Ca^2+^ affinity of the EF-hand of C18S Ocm, which is filled first, is 1.5 orders of magnitude lower than that of rWT Ocm (Table 4). As the CD and EF sites of Ocm are considered as low- and high-affinity sites, respectively [67], the C18S substitution seems to suppress Ca^2+^ affinity of the EF site.

Mg^2+^-binding to rWT/C18S Ocm was studied by equilibrium dialysis method with determination of total Mg^2+^ concentration using electrothermal atomization atomic absorption spectrometry (Figure 5). Scatchard plots (Figure 5b) reveal two linear regions corresponding to filling by Mg^2+^ of the two metal-binding sites. Again, analysis of the data within the sequential metal-binding scheme [IV] evidences that our estimates of Mg^2+^ affinity of rWT Ocm (Table 4) are 0.9–1.6 orders of magnitude higher compared to those reported earlier [66]. Similar to the results for Ca^2+^-binding to Ocm, the C18S substitution suppresses Mg^2+^ binding to the site, which is occupied first, by 0.4 orders of magnitude (Table 4). Importantly, the ratio of total free energies of Mg^2+^ and Ca^2+^ binding to C18S Ocm (calculated as Δ*G_∑_* = −RT∙(ln *K_a1_* + ln *K_a2_*)) equals to the same ratio calculated for rWT Ocm (0.64; Appendix A). Therefore, the C18S substitution equally compresses the energetic scales of Mg^2+^ and Ca^2+^ binding to Ocm, thereby confirming self-consistency of our estimates of the metal-binding constants.

In accordance with the lowered Ca^2+^ affinity of C18S Ocm, fluorimetric stopped-flow measurements of EGTA-induced Ca^2+^ removal from the protein (Figure 4b) demonstrate accelerated compared to rWT Ocm Ca^2+^ dissociation from the mutant. Analysis of the kinetic data according to the sequential metal-binding scheme [IV] (Table 5) shows that Ca^2+^ dissociation from the site of C18S Ocm, which is occupied first, occurs 38% faster than from that of rWT Ocm. Ca^2+^ dissociation from the second site of C18S Ocm occurs within the dead time of the instrument (6 ms). In contrast to the dissociation stage, Ca^2+^ association with the first site of the C18S mutant is 20-fold slower in comparison to rWT Ocm (Table 5). Thus, Ca^2+^ affinity of C18S Ocm is decreased mostly due to the drop in Ca^2+^ association rate. Mg^2+^ dissociation from rWT/C18S Ocm occurs within the dead time of the instrument (data not shown), in accordance with their relatively low Mg^2+^ affinities (Table 4).

Overall, the structural and functional consequences of the homologous C18S substitution are unexpectedly pronounced. For this reason, the numerous possible oxidative modifications of the C18 thiol could induce noticeable changes in functioning of Ocm. Similarly, the oxidation mimicking substitution of the conservative C39 residue of recoverin markedly changes its structural properties and affinity for photoreceptor membranes [68].

## 4. Conclusions

We have shown that the highly conservative C18 thiol of rat Ocm exhibits physiologically relevant redox potential values, which suggests its disulfide dimerization under oxidizing conditions, inherent to numerous disorders. Another possibility in this case is an accumulation of the thiol-oxidized derivatives. As minor structural modification of the C18 residue (C18S substitution) significantly affects structural and functional status of Ocm, its thiol derivatives are expected to possess altered functional activity. Thus, oxidizing conditions are likely to favor accumulation of the oxidized forms of Ocm with modified activity, thereby supporting the redox sensory function of its C18 thiol. The effects described in the present work closely resemble those reported for the conservative among neuronal calcium sensor proteins C39 residue of recoverin, located in the loop of its inactive EF-hand 1. Light-induced oxidation of this thiol results in appearance under in vivo conditions of disulfide dimer of recoverin and its oxidized monomer, which demonstrate altered structural and functional properties [41,63]. Ocm and recoverin are both descendants of calmodulin [69,70], but their conservative thiols located in the inactive EF-loops corresponding to EF-hands 2 and 1 of calmodulin, respectively. Therefore, the selected EF-hands of Ocm and recoverin independently lost affinity to calcium in the course of their evolution, at the expense of the gain in redox sensitivity. The appearance of redox sensitivity in calcium-binding proteins gives them a unique ability to serve as a hub for calcium signaling and redox signaling pathways. The role of redox-sensitive EF-hand proteins in the well-studied crosstalk between calcium and reactive oxygen species [71,72] needs further clarification.

## Figures and Tables

**Figure 1 biomolecules-11-00066-f001:**
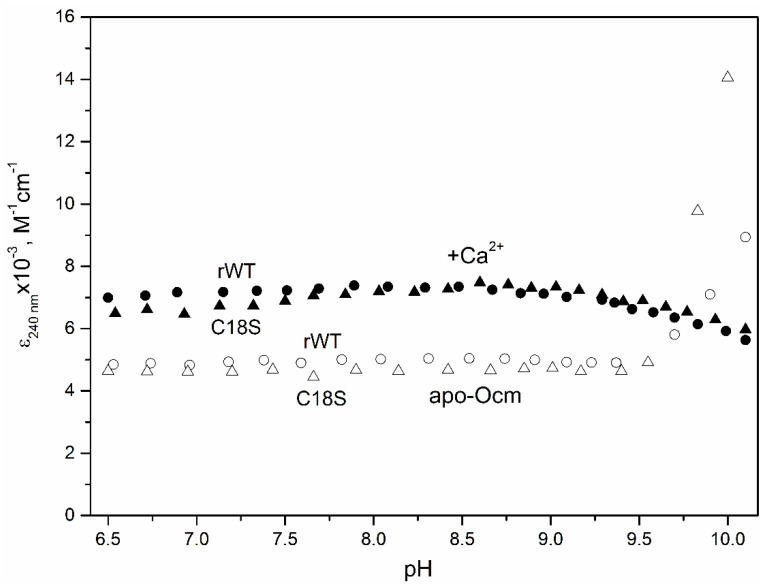
pH dependence of molar extinction coefficient at 240 nm at 20 °C, after subtraction of contribution of the deprotonated Tyr residues, for apo- (open symbols) and Ca^2+^-loaded (solid symbols) forms of reduced rWT Ocm (circles) and C18S Ocm (triangles). Protein concentration was 23 μM. Buffer conditions: 30 mM MES, 30 mM HEPES, 30 mM H_3_BO_3_, 30 mM glycine, 100 mM KCl, 1.5 mM EDTA (apo-Ocm), or 1 mM CaCl_2_ (Ca^2+^-bound Ocm).

**Figure 2 biomolecules-11-00066-f002:**
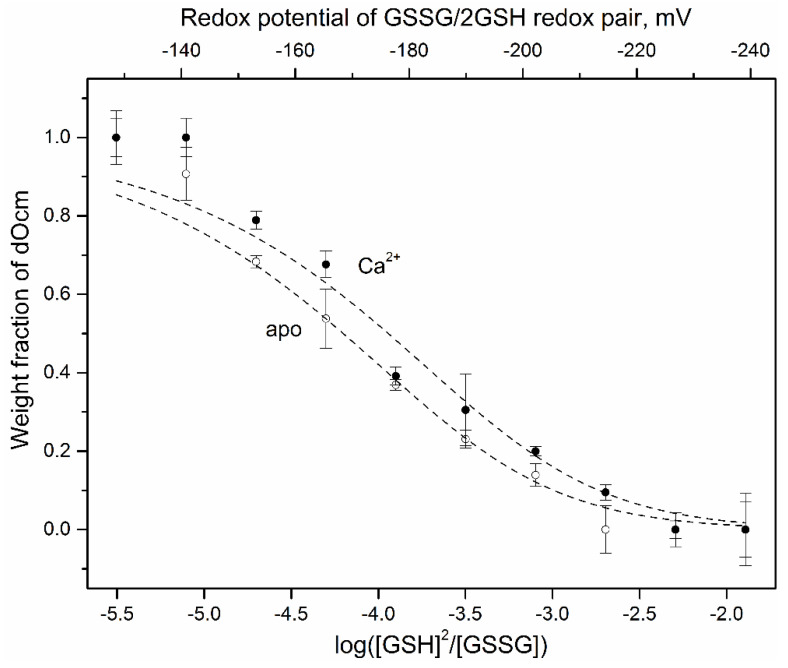
Quantitation of redox equilibrium between apo-/Ca^2+^-loaded dOcm and GSSG/2GSH redox couple at 35 °C. 1.5 μM dOcm solution (50 mM HEPES-KOH, 100 mM KCl, 1 mM EDTA/CaCl_2_, pH 7.4) was incubated under anaerobic conditions with 0.2 mM GSSG and 25 μM to 1.6 mM GSH for 62 h, followed by non-reducing SDS-PAGE. The weight fractions of dOcm were derived from quantitative analysis of the gels. The top axis corresponds to the redox potential of GSSG/2GSH redox pair (scheme [II]) calculated according to Equation (4). The dashed curves correspond to theoretical fits of the data computed according to scheme [I] using Equation (1).

**Figure 3 biomolecules-11-00066-f003:**
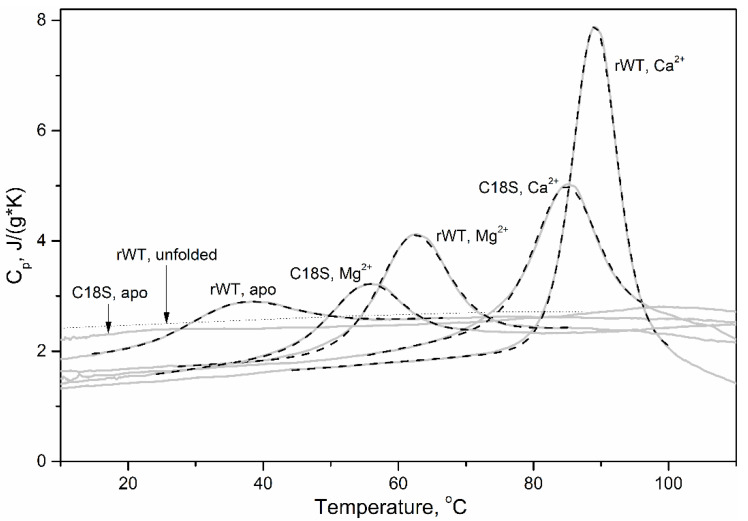
Temperature dependencies of specific heat capacities of apo- and metal-bound states of rWT/C18S Ocm, derived from DSC data [46]. Protein concentration was 1.2–2.1 mg/mL. Buffer conditions: 10 mM H_3_BO_3_-KOH, 1 mM CaCl_2_, pH 9.0 (Ca^2+^-bound Ocm), 20 mM glycine-KOH, pH 9.2, 1 mM EDTA (apo-Ocm), or 20 mM glycine-KOH, pH 9.2, 1 mM MgCl_2_ (Mg^2+^-bound Ocm). Heating rate was 1 K/min. The DSC data are analyzed according to the cooperative two-state model [III]. The experimental and theoretical curves are shown solid and dashed, respectively. The dotted curve corresponds to specific heat capacity of the fully unfolded rWT Ocm, calculated according to ref. [48].

**Figure 4 biomolecules-11-00066-f004:**
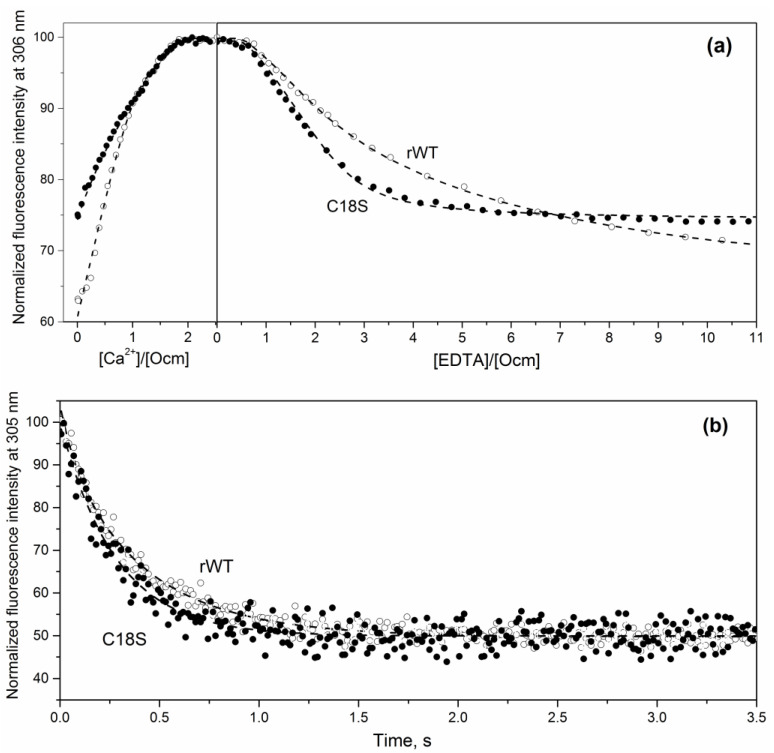
Fluorimetric Ca^2+^ and EDTA titration of rWT/C18S Ocm (**a**) and time course of EGTA-induced Ca^2+^ removal from Ocm monitored by fluorescence stopped-flow technique (**b**) at 20 °C. Buffer conditions: 10 mM HEPES-KOH, pH 8.2. Protein concentration was 5–23 µM. Excitation wavelength was 275 nm. The dashed curves are theoretical fits computed according to the sequential metal-binding scheme [IV] (see Table 4 and Table 5). (**a**) The dependence of Ocm fluorescence intensity at 306 nm on total concentration of Ca^2+^/EDTA during saturation of apo-Ocm with Ca^2+^, followed by EGTA-induced Ca^2+^ removal. (**b**) Ca^2+^ to protein molar was 2:1; the Ca^2+^ removal was initiated via addition of 1 mM EGTA.

**Figure 5 biomolecules-11-00066-f005:**
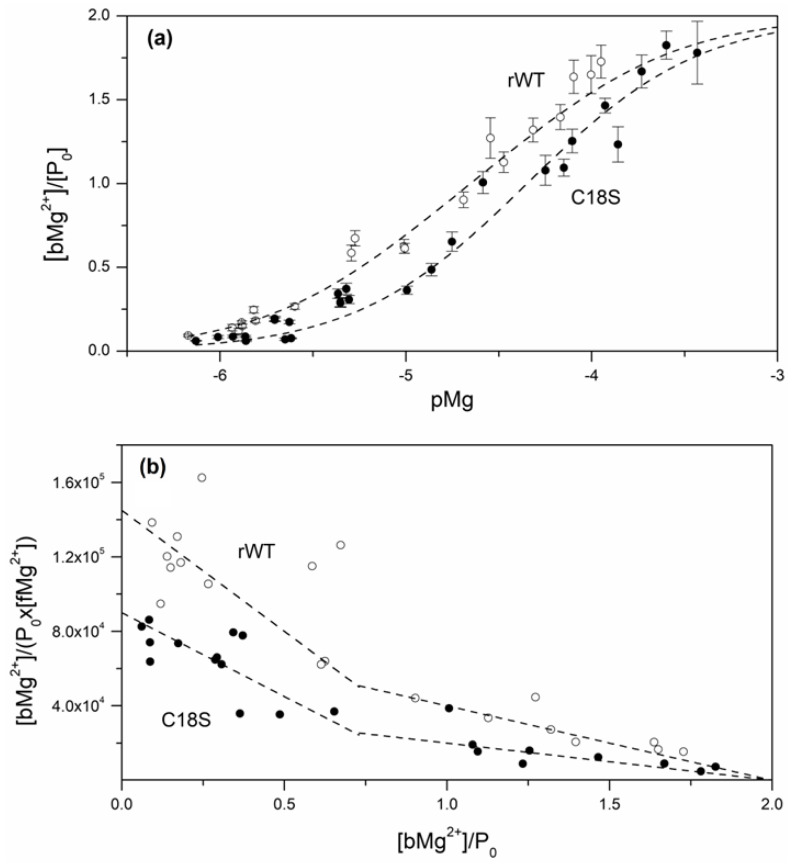
Determination of Mg^2+^ affinity of rWT/C18S Ocm at 20 °C by equilibrium dialysis. Ocm concentration was 50–200 μM. Buffer conditions: 30 mM HEPES-KOH, 1 mM EGTA, pH 7.4 and 2–750 μM Mg(NO_3_)_2_. [bMg^2+^], concentration of Mg^2+^ bound to Ocm; [fMg^2+^], free Mg^2+^ concentration; P_0_, total Ocm concentration. (**a**) Dependence of the number of Mg^2+^ ions bound per Ocm molecule upon pMg. The points are experimental, dashed curves are theoretical computed according to the sequential metal binding scheme [IV] using Equation (5) (see Table 4). (**b**) Scatchard plot.

**Table 1 biomolecules-11-00066-t001:** The secondary structure fractions of rWT/C18S Ocm (7–13 µM) estimated from far-UV CD spectra [45] and parameters of fluorescence emission spectrum (*λ_max_*, spectrum maximum position; *I_max_*, maximum fluorescence intensity) of hydrophobic probe bis-ANS (1 µM) in presence of the protein (18–23 µM) at 20 °C. Buffer conditions: 10 mM H_3_BO_3_-KOH, pH 8.8 or 10 mM Tris-HCl, pH 7.3 for CD studies; 10 mM HEPES-KOH, pH 7.3/8.2 for bis-ANS fluorescence measurements. Ca^2+^/Mg^2+^ content was controlled by addition to the metal-depleted protein of 1.5 mM EDTA (apo-Ocm), 1 mM CaCl_2_ (Ca^2+^-loaded Ocm) or 1 mM EGTA, 1 mM MgCl_2_ (Mg^2+^-loaded Ocm). Standard deviations for estimates of the secondary structure fractions are indicated.

Ocm State	Secondary Structure Fractions				Bis-ANS Fluorescence Emission					
	rWT		C18S		rWT		C18S		Free Bis-ANS	
	α-helices,%	β-sheets, %	α-helices, %	β-sheets, %	*λ*_max_, nm	*I*_max_, a.u.	*λ*_max_, nm	*I*_max_, a.u.	*λ*_max_, nm	*I*_max_, a.u.
apo	32.4 ± 4.3	17.1 ± 3.0	12.7 ± 1.8	26.3 ± 2.3	496	90	497	115	538	8
Mg^2+^	53.2 ± 4.3	9.1 ± 3.0	32.6 ± 1.8	15.9 ± 2.3	494	74	496	98	539	8
Ca^2+^	54.8 ± 4.3	8.0 ± 3.0	53.9 ± 1.8	8.1 ± 2.3	499	34	494	58	536	8

**Table 2 biomolecules-11-00066-t002:** Study of quaternary structure of rWT/C18S Ocm at 20 °C according to chemical cross-linking with 0.02% glutaric aldehyde and DLS data (mean hydrodynamic radius, *R_h_*); 1.3–1.7 mg/mL Ocm in 20 mM H_3_BO_3_, pH 8.4 buffer, 1.5 mM EDTA or 1 mM MgCl_2_/CaCl_2_ for the apo- and Mg^2+^/Ca^2+^-loaded protein, respectively. Standard deviations for the *R_h_* values are indicated.

Ocm	Protein State	Fractions of Protein Multimers		*R_h_*, nm
		12 kDa Band, %	24 kDa Band, %	
rWT	apo	100	0	2.22 ± 0.08
C18S		100	0	2.42 ± 0.04
rWT	Mg^2+^	100	0	2.01 ± 0.06
C18S		100	0	1.97 ± 0.04
rWT	Ca^2+^	96	4	1.80 ± 0.03
C18S		95	5	1.81 ± 0.04

**Table 3 biomolecules-11-00066-t003:** Thermodynamic parameters of thermal denaturation of various states of rWT/C18S Ocm, estimated from the DSC data shown in Figure 3 according to the cooperative two-state model [III]: mid-transition temperature (*T*_0_), enthalpy of protein denaturation at temperature *T*_0_ (∆*H*_0_), heat capacity change accompanying the transition (∆*C_p_*), and number of molecules involved in the transition (*n*).

Ocm	Protein State	*T*_0_, °C	Δ*H*_0_, J/g	Δ*C_p_*, J/(g·K)	*n*
rWT	apo	30.7	11.3	0.59	0.83
	Mg^2+^	60.9	27.2	0.26	0.83
	Ca^2+^	89.3	52.9	−0.31	0.77
C18S	apo	n.d.	n.d.	n.d.	n.d.
	Mg^2+^	55.9	16.4	−0.09	1.25
	Ca^2+^	85.6	31.8	−0.41	0.87

n.d.: not determined.

**Table 4 biomolecules-11-00066-t004:** The values of equilibrium Ca^2+^ association constants for rWT/C18S Ocm at 20 °C estimated from the experimental data shown in Figure 4a (fluorimetric Ca^2+^/EDTA titrations) and 5 (equilibrium dialysis) according to the sequential metal-binding scheme [IV]. Standard deviations for the estimates of equilibrium metal binding constants (*K_a_*_1_ and *K_a_*_2_) are indicated.

Ca^2+^			Mg^2+^	
Ocm	Fluorimetric Ca^2+^/EDTA Titrations		Equilibrium Dialysis	
	log *K_a_*_1_	log *K_a_*_2_	log *K_a_*_1_	log *K_a_*_2_
rWT	9.1 ± 0.3	7.5 ± 0.3	5.15 ± 0.03	4.16 ± 0.09
C18S	7.6 ± 0.2	7.9 ± 0.2	4.71 ± 0.04	4.00 ± 0.19

**Table 5 biomolecules-11-00066-t005:** Equilibrium and kinetic parameters of Ca^2+^ binding to rWT/C18S Ocm at 20 °C, estimated from the fluorimetric experiments. The values of equilibrium metal association constants, *K_a_*_1_ and *K_a_*_2_, and rate constants of metal dissociation, *k_off_*_1_ and *k_off_*_2_, were estimated according to the sequential metal-binding scheme [IV] from equilibrium and stopped-flow experiments shown in Figure 4a,b, respectively. The kinetic constants of Ca^2+^ binding to Ocm, *k_on_*_1_ and *k_on_*_2_, were derived from these estimates. Standard deviations for the parameters of Ca^2+^ binding to Ocm are indicated.

Ocm	*K_a_*_1_, M^−1^	*K_a_*_2_, M^−1^	*k_on_*_1_, M^−1^s^−1^	*k_on_*_2_, M^−1^s^−1^	*k_off_*_1_, s^−1^	*k_off_*_2_, s^−1^
rWT	(1.3 ± 0.9) × 10^9^	(4 ± 2) × 10^7^	(3 ± 2) × 10^9^	(5 ± 4) × 10^8^	2.6 ± 0.3	14.4 ± 5.0
C18S	(4 ± 2) × 10^7^	(8 ± 4) × 10^7^	(1.5 ± 0.8) × 10^8^ *	n.d. *	3.6 ± 0.4 *	n.d. *

* Ca^2+^ dissociation from the second site occurs within dead time of the instrument (6 ms).

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
