# Peer review of "The Highly Conservative Cysteine of Oncomodulin as a Feasible Redox Sensor"

_biomolecules, 2021, doi:10.3390/biom11010066_

Round 1
Reviewer 1 Report
Dear Editor,
The manuscript entitled “The highly conservative cysteine of oncomodulin as a feasible redox sensor” by Vologzhannikova et al. is trying to elucidate the function of a conserved Cys18 of oncomodulin. For that purpose, the authors have studied the influence of a homologous C18S substitution, on thiol oxidation, conformational stability and metal-binding properties of a recombinant rat Ocm. The authors conclude that the presented data favor the view that the highly conservative Cys18 of rat Ocm likely serves as a redox sensor for the cell.
In my opinion, the manuscripts’ objective and findings are interesting, the study is well designed and well explained and the manuscript is well-written. Therefore I think it should be accepted for publication after minor revisions. My detailed comments for the authors to consider are provided below:
- The ocm and a-PA comparison paragraph in the introduction section should be shortened.
- In section 2.7, equation (1) is confusing for me. Further explanation or a specific reference might help.
- In section 2.15, I would like to know why the sequential metal binding scheme was chosen, as the best theoretical model.
- Section 3.2, is a bit confusing. I think it should be rephrased for more clarity.
- References in German or Russian should be replaced with references in English to be easier to read for a wider readership.
- I might be wrong since I’m not a native English speaker but conservative Cys shouldn’t be conserved Cys throughout the text?
Overall, it is a very well written and informative manuscript.
Author Response
1. We have condensed this paragraph removing some information about a-PA.
2. We have tried to give some explanation how to obtain equation (1) on page 5. The mathematics
is very simple and we do not think it is worth to give more details.
3. In our previous works and in the works of many other authors it was shown that calcium and
magnesium binding to parvabumins proceeds sequentially. It was discussed for example in [2].
For this reason, we have used this scheme of the binding in our work. We have provided the
corresponding reference to the text in sections 2.14 and 2.15.
4. We have tried to make section 3.2 more clear
5. We did not find references in Russian and translated into English the only German reference
6. Conserved is changed for conservative.
Reviewer 2 Report
This is an extensive, will carry it out study by first rate investigators in the field of proteins structure and conformation. There are a few somewhat minor concerns about the experiments and their significance.
- One needs to be careful when measuring file oxidation is this can be influenced by trace transition metal ions and oxygen.
- Is dimerization physiologically significant?
- It is somewhat possible to characterize the theology oxidation states, ie., the oxidized chemistry: has that been done and is it relevant?
- this reviewer’s opinion is that the significance of the thiol oxidation might be less important than the structural perturbations between replacement of this group
- Hence a more speculative, but qualified discussion is in order
In summary, lots of detailed accurate experimental data.
Author Response
1. In Materials and Methods:
All buffers and other solutions were prepared using ultrapure water (Millipore Simplicity 185 system). Degassed buffers were used for preparation of DTT solutions just before to their usage to avoid DTT oxidation. Quantitation of thiol groups of rWT Ocm (1 Cys residue) was performed according to the modified Ellman’s test [36] .
2. It is unknown. Physiological role of parvalbumins in many systems is unknown.
3. Sorry, but we do not understand what is ‘theology oxidation states’…
Section 3.4: Aside from disulfide dimerization of Ocm, oxidizing physiological conditions may induce numerous chemical modifications of its Cys18 residue, including formation of mixed disulfides, sulfenic, sulfinic and sulfonic acids [64] , as observed for recoverin [63] .
4. We think that the structural perturbations in these two cases are comparable. SH vs. OH is the only difference.